# Diversity of mosquitoes (Diptera: Culicidae) collected in different types of larvitraps in an Amazon rural settlement

Jessica Feijó Almeida[1,2,3] *, Heliana Christy Matos Belchior[1,4], Claudia María Ríos-Velásquez[1,2,4], Felipe Arley Costa Pessoa[1,2,3,4] *

1 Laboratório de Ecologia e Doenças Transmissíveis na Amazônia, Instituto Leônidas e Maria Deane—Fiocruz Amazônia, Manaus, Amazonas, Brasil, 2 Programa de Pós-Graduação em Condições de Vida e Situações de Saúde na Amazônia, Instituto Leônidas e Maria Deane—Fiocruz Amazônia, Manaus, Amazonas, Brasil, 3 Programa de Pós-Graduação em Entomologia, Instituto Nacional de Pesquisas da Amazônia, Manaus, Amazonas, Brasil, 4 Programa de Pós-Graduação em Biologia da Interação Patógeno-Hospedeiro, Instituto Leônidas e Maria Deane—Fiocruz Amazônia, Manaus, Amazonas, Brasil

☉ These authors contributed equally to this work.
* jessicalmeida1993@gmail.com (JFA); facpessoa@gmail.com (FACP)

**Data Availability Statement:** All relevant data are within the manuscript and its Supporting Information files.

## Abstract

Anthropogenic environments provide favorable conditions for some species, which is especially true of mosquitoes that present eclecticism at the moment of choice for the site of oviposition. In the present study, the diversity of mosquitoes was assessed by providing plastic containers, bamboo internodes, and tires in a forest, the forest edge, and peridomicile environments in a rural settlement area. Eighteen sampling points were chosen, delimited by a buffer of 200 m, placed in three environments: forest, forest edge, and peridomicile. In each environment, larvitraps were installed, separated by a minimum distance of 7 m and 1 m from the ground. A total of 10,131 immature mosquitoes of 20 species were collected. The most abundant species was *Culex urichii* (29.5%), followed by *Trichoprosopon digitatum* (27.1%), and *Cx.* (*Melanoconion*) spp. (10.4%). There was a difference in the composition of immature mosquito populations between larvitraps (p < 0.0005), and the plastic container hosted a greater diversity of species, whereas tires presented a greater abundance of individuals. The forest, forest edge, and peridomicile environments were also different with regard to diversity of immature mosquito populations (p < 0.0010). The forest edge was the environment with the greatest diversity of species, followed by the peridomicile and forest environments. In the forest and peridomicile, plastic container larvitraps had the greatest diversity, whereas the forest edge tire presented the largest number of individuals. Further, tire larvitraps collected the largest number of individuals in all environments. Ten species associated with the bamboo internode and tire were identified. The preference of species for artificial larvitraps, such as the plastic container and tire, even in wild environments was noted. These artificial objects may represent a risk factor for the population living in this region, as all vector species found in the study were present in plastic containers and tires.

**Funding:** This study was supported by Fundanção de Amparo a Pesquisa do Estado do Amazonas – Programa de Excelência em Pesquisa Básica e Aplicada em Saúde in the form of a grant awarded to FACP (1990/2013-FAPEAM). The article is a by-product of the umbrella project "Arboviroses emerging in the Amazon: Alphavirus incidence risk factors with an emphasis on Mayaro, on the agricultural frontier in Central Amazon ", under the coordination of FACP. Information for this grant can be found at http://www.fapeam.am.gov.br/wp-content/uploads/2014/07/DecCD-159_2014.pdf. This study was also supported by Instituto Leônidas e Maria Deane - Fiocruz Amazônia in the form of a Scientific Initiation Scholarship awarded to HCMB (PAIC 00812016/2016), Fundação Oswaldo Cruz in the form of a scholarship awarded to JFA (Scholarship Fiocruz/VPEIC/2016-2018), and Fundação de Amparo à Pesquisa do Estado do Amazonas - Programa de Apoio à Publicação de Artigos Científicos in the form of a grant (01.01.016301.0000506.2019-FAPEAM). The funders had no role in study design, data collection and analysis, decision to publish, or preparation of the manuscript.

**Competing interests:** The authors have declared that no competing interests exist.

## Introduction

Insects are the most diverse of all animal classes on the planet, and Brazil is the country with the greatest insect diversity, with estimates of 400–500 thousand known species, with most of these inhabiting the Amazon forest [1–3]. Anthropic activity has affected some insect species, especially dipteran vectors such as sand flies, biting midges, and mosquito populations [4–6]. Mosquitoes are vulnerable to changes in environment and climate caused by deforestation and land use. Environmental changes affect the distribution of Culicidae, leading to the increased abundance of some species and a decreased abundance of others. Inevitably, the dynamics of disease transmission by mosquitoes are also affected [7, 8].

Anthropogenic changes in the forest environments caused by rural settlements become favorable for human-vector interaction due to their proximity to extensive forest areas and the lack of adequate infrastructure for the residents who live there. In the Amazon region, precarious assistance by the government in rural settlements and the traditional practices e.g. hunting, extraction of medicinal herbs and sylvatic fruits have led to an increase in cases of malaria and arboviruses [9, 10].

A study carried out in a settlement in Amazonas, Brazil found a high seroprevalence of the *Mayaro* virus in residents, including people who did not enter the forest, such as children and women, suggesting that this arbovirus is also transmitted by species other than the main vector, *Haemagogus janthinomys* Dyar [11]. It was also observed that, in the settlement, adult *Ochlerotatus serratus* (Theobald), *Psorophora cingulata* (Fabricius), *Hg. tropicalis* Cerqueira and Antunes mosquitoes had been naturally infected with the *Oropouche* virus, in addition to the typical acrodendrophilous species captured in the soil. Further, a greater diversity of species was reported in the forest edge environment when compared to the forest and peridomicile [12].

In general, mosquito species exhibit a specialization in oviposition site selection, while others are opportunistic with respect to those behaviors, ranging from small and ephemera to large and permanent [13, 14]. Man-made objects are also perceived as potential breeding sites for mosquitoes [15]. Immature *Limatus durhamii* Theobald mosquitoes, a species naturally infected with the *Guama* virus, can develop in breeding sites ranging from tree holes to landfill percolation tanks [16, 17]. Immature forms of *Anopheles* species have been recorded in artificial habitats [18, 19], including *An. darlingi* Root, which was found in an artificial lagoon in the urban region of Manaus in the state of Amazonas [20].

Studies of immature mosquitoes captured in artificial breeding sites located in Amazonian environments are scarce. However, some authors [15, 21–25] have indicated that artificial breeding sites are preferred by immature mosquitoes of various species.

Estimating the diversity of mosquito species found in artificial habitats gives us an indication of which species are adapted and/or are opportunistic to these types of habitats. Therefore, this work aimed to identify mosquito species that colonize different types of artificial breeding places, located in environments with different levels of anthropization within the rural settlement state of Amazonas.

## Materials and methods

### Study site and sample design

The study was carried out in the Rio Pardo rural settlement, Presidente Figueiredo Municipality, State of Amazonas, Brazil (01º49'02.4" S, 060º19'03.6" W) and borders the Canoas settlement, the Waimiri-Atroari indigenous reserve and private land, and the Brazilian Federal Government (Fig 1). The settlement is surrounded by a continuous primary forest, presenting

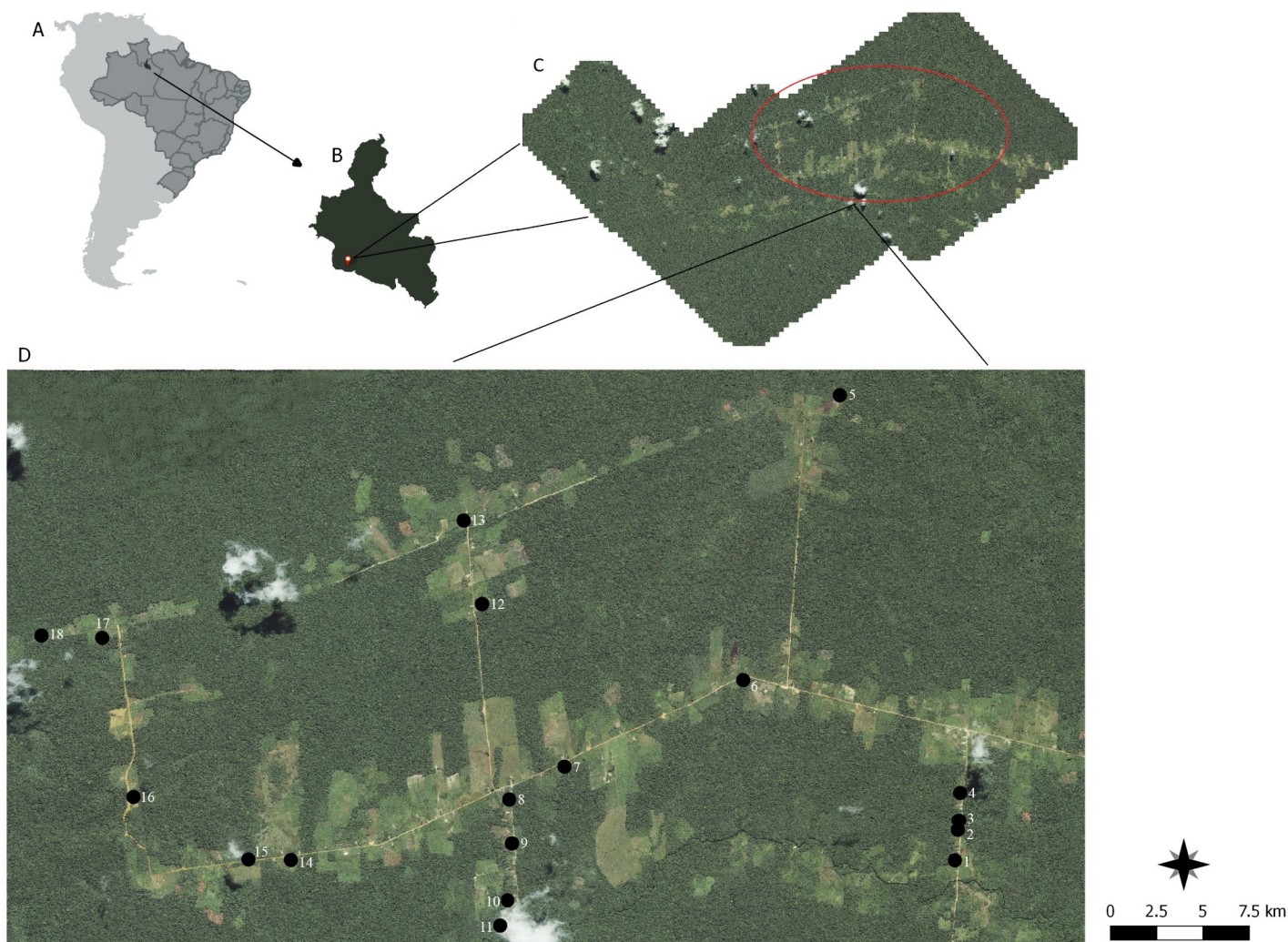

**Fig 1. Study area geographical information system. A**—In different shades of grey; South America, Brazil, and the municipality of Presidente Figueiredo. **B**—Delimitation area of the Presidente Figueiredo municipality, pointing to the agrovillage of Rio Pardo. **C**—Rio Pardo agrovillage and, in particular, roads where larvitraps were distributed for immature mosquito collection. **D**—Location of collection points along agrovillage roads: the numbers represent the geographic coordinates of the collection points: 1 (01º48'37.6" S, 060º16'27.1" W); 2 (01º48'26.7" S, 060º16'26.1" W); 3 (01º48'23.5" S, 060º16'25.7" W); 4 (01º48'13.6" S, 060º16'25.2" W); 5 (01º45'51.6" S, 060º17'08.2" W); 6 (01º47'33.3" S, 060º17'42.7" W); 7 (01º48'30.4" S, 060º18'46.8" W); 8 (01˚48'15.9" S, 060˚19'06.3" W); 9 (01˚48'31.6" S, 060˚19'05.3" W); 10 (01˚48'51.9" S, 060˚19'06.9" W); 11 (01˚49'00.9" S, 060˚19'09.5" W); 12 (01˚47'06.1" S, 060˚19'15.9" W); 13 (01˚46'36.3" S, 060˚19'22.5" W); 14 (01º48'37.5" S, 060º20'24.2" W); 15 (01º48'37.3" S, 060º20'39.4" W); 16 (01º48'15.0" S, 060º21'20.4" W); 17 (01˚47'18.2" S, 060˚21'31.6" W); 18 (01˚47'17.4" S, 060˚21'53.3" W). Reprinted from Maxar under a CC BY license, with permission from Instituto Leônidas e Maria Deane—Fiocruz, original copyright 2008.

a humid tropical climate according to the Köpper classification, with an average temperature and annual rainfall of 27.1 ºC and 2975 mm, respectively, and two climatic seasons; drought from June to October and rainy from November to May [26]. The human population of the region is comprised of approximately 550 inhabitants, and economic activities include agriculture and livestock.

The area of the settlement is about 317 km$^2$, and includes roads, small villages, wooden houses, gardens, and forest areas. The deforestation rate of the settlement increased from 12.32% in 2008 to 19.79% in 2015 (FRF, unpublished data).

The delimitation of the settlement area was done through IKONOS$^{TM}$ satellite imagery. The sampling points were selected by presence of a dwelling with peridomicile, forest edge,

and forest environment in a 200 m buffer. Here, the peridomicile environment is defined as the area with garden, fruit garden, and animal shelters located around the dwelling; the forest edge is the transitional area between the peridomicile and forest environment; and, the forest area is characterized by preserved forest. To calculate the number of sampling points, 120 available sampling points were detected from satellite images (PRODES®, 1 m x 1 m, August 2008), and then, 18 of them were selected using software RStudio program [27].

## Collection of immature mosquitoes

Collections were carried out during four periods of 15 days each, during the months of November 2017 and January and February 2018. For the collection of immature mosquitoes, bamboo internodes, tires, and plastic containers were installed (Fig 2). Every larvitrap was carefully washed before installation to avoid contamination between them. One unit of each larvitrap type was installed in the forest, forest edge, and peridomicile environments, separated

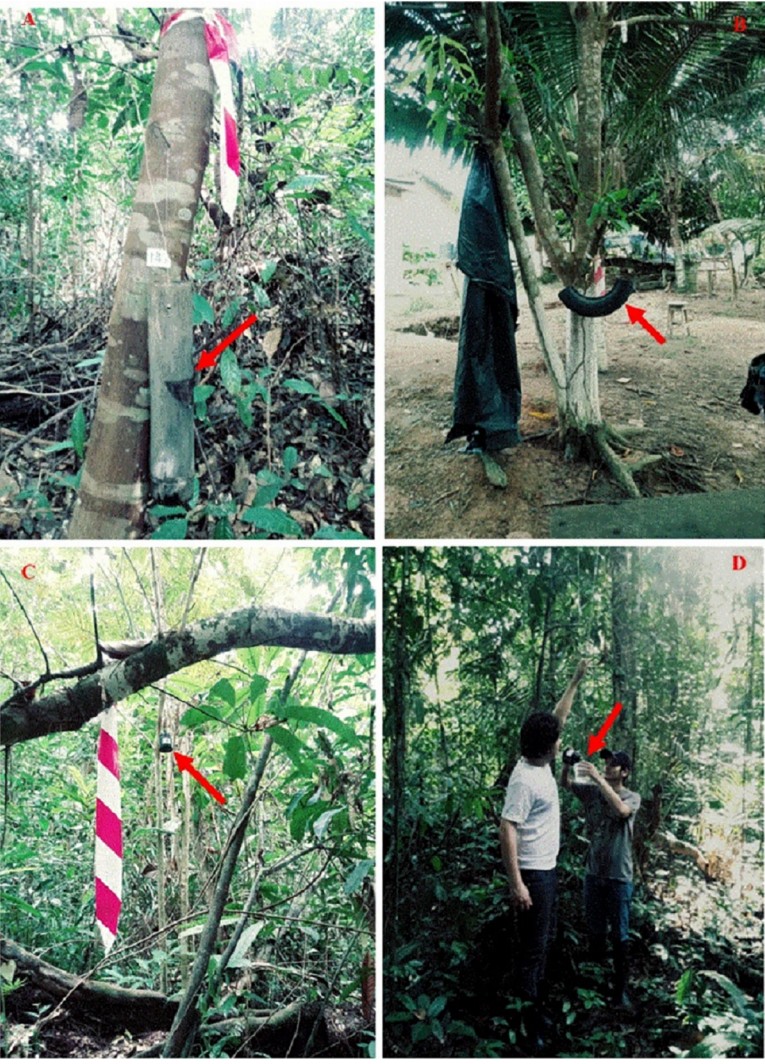

**Fig 2. Larvitrap installations in the Rio Pardo agrovillage, Presidente Figueiredo, Amazonas, Brazil. A**—bamboo internode. **B**—tire. **C**—plastic container. **D**—trap installation. The arrows point to the installed larvitraps for the collection of immature mosquitoes.

by at least 7 m from each other trap and 1 m above ground level. Nine larvitrap sets were installed at each sampling point area with 162 total traps for each collection event. Traps were filled with 500 ml of untreated well water. The larvitraps were installed and they were not removed until the end of all collection periods. Approximately forty larvitraps were installed per day, over four days.

The collection of immature mosquitoes began 15 days after installation of the traps, which is a regular time for larval development for the most Neotropical mosquito species [27]. Inside the larvitraps was a clear plastic polyethylene bag carefully closed with a rubber band allowing air to stay inside for larval respiration. Every plastic bag was labelled. Larvitraps were then filled with water again. Collected larvae were sorted to separate them from occasional predators and were reared in plastic containers containing water and were fed with organic material obtained from the larvitraps at the time of collection until they reached the fourth instar or adult stage. The permanent mosquito collection license belongs to FACP, nº 12186, issued by SISBIO-ICMBio. The individual in this manuscript has given written informed consent (as outlined in PLOS consent form) to publish these case details (Fig 2). The project is authorized by the Research Ethics Committee Nº CAAE 40108114.7.0000.5248 and permission issued by the community president, Mr. Eleonildes Fernandes da Silva.

Mosquitoes were identified using the identification keys of Lane [28, 29], Forattini [27], Consoli and Lourenço-de-Oliveira [29], and Zavortink [30]. Genera were abbreviated according to Reinert [31, 32].

The vouchers were deposited in the collection of the Instituto Leônidas e Maria Deane/ Laboratório de Ecologia e Doenças Transmissíveis da Amazônia (ILMD—LEDTA), Fiocruz Amazônia.

## Statistical analysis

To evaluate sampling, the rarefaction curve of the identified species was used as a function of the frequency of individuals captured. Shannon-Wiener and Simpson diversity indices, Berger-Parker dominance index, Pielou equitability, and Jaccard similarity were used to analyze the diversity patterns and species distribution between the larvitraps and environments.

To verify the influence of different environments and trap types on mosquito population composition, an Analysis of Permutational Multivariate Variance (PERMANOVA) was performed, which consisted of multivariate non-parametric analyses based on permutations. The Indication Value of Species (IndVal) was applied to verify the specificity and fidelity of the mosquito species to the larvitraps analyzed.

Statistical analyses were carried out using the program Past Version 3.14 [33] and the free statistical software RStudio Version 1.2.1335 [34] with the 'vegan' [35], 'labdsv' [36], 'ggplot2' [37] and 'indicspecies' [38] packages.

## Results

In total, 10,868 larvae were collected, and 7% of them were not identified because they died during transportation from the field to the laboratory. A total of 10,131 immature mosquitoes were collected, grouped into 10 genera and 20 species. The most abundant species were *Culex urichii* Coquillett, with 2,988 individuals (29.5%), *Trichoprosopom digitatum* Theobald, with 2,746 (27.1%), and *Cx.* (*Melanoconion*) spp. with 1,052 (10.4%).

The tire was the most effective larvitrap type, with the highest number of individuals, 6,195 (61.1%), followed by the bamboo internode with 2,593 (25.5%) and the plastic container with 1,343 (13.2%). The bamboo internode collected the largest number of species with a total of 17 species, followed by the plastic container with 16 and the tire with 15.

Among the mosquito species collected, *Ochlerotatus argyrothorax* Bonne-Wepster and Bonne was frequently found only in tire larvitraps, *Sabethes cyaneus* Fabricus in the plastic containers, while *Orthopomyia fascipes* Coquillett, *Sa. amazonicus* Gordon and Evans, and *Sa. belisarioi* Neiva were frequently observed in bamboo internode traps. All other species were frequently recorded in at least two larvitrap types (Table 1).

The rarefaction curve of accumulated species richness revealed that plastic container and tire curves had asymptotes, with curves stabilizing around 1,100 individuals and 16 species and 3,500 individuals and 15 species, respectively, in the bamboo internode trap a trend toward stabilization of curves was observed (Fig 3).

The plastic container larvitrap presented greater species diversity and equitability of immature mosquitoes when compared to the bamboo internode and tire (Table 2, Fig 3). The greatest similarity in the diversity of immature mosquitoes was observed between the plastic container and the tire (Cj = 0.82).

**Table 1. Analysis of captured immature mosquitoes.**

| Species | Forest | | | | Forest edge | | | | Peridomicile | | | |
|---|---|---|---|---|---|---|---|---|---|---|---|---|
| | Bamboo internode* | Plastic containers* | Tire* | Total | Bamboo internode | Plastic containers | Tire | Total | Bamboo internode | Plastic containers | Tire | Total |
| *Aedes albopictus* (Skuse) | 0 | 6 | 38 | 44 | 0 | 1 | 48 | 49 | 80 | 125 | 279 | 484 |
| *Culex* (*Melanoconion*) spp. | 175 | 199 | 126 | 500 | 39 | 123 | 203 | 365 | 26 | 112 | 49 | 187 |
| *Cx. nigripalpus* Theobald | 71 | 15 | 135 | 221 | 35 | 9 | 120 | 164 | 38 | 30 | 105 | 173 |
| *Cx. quinquefasciatus* Say | 0 | 0 | 0 | 0 | 0 | 1 | 3 | 4 | 0 | 0 | 0 | 0 |
| *Cx. urichii* (Coquillett) | 11 | 47 | 1,731 | 1,789 | 2 | 40 | 626 | 668 | 71 | 15 | 445 | 531 |
| *Haemagogus janthinomys* Dyar | 4 | 1 | 0 | 5 | 5 | 4 | 1 | 10 | 18 | 12 | 0 | 30 |
| *Limatus durhamii* Theobald | 6 | 4 | 104 | 114 | 4 | 71 | 374 | 449 | 109 | 162 | 113 | 384 |
| *Li. flavisetosus* Oliveira Castro | 0 | 19 | 122 | 141 | 0 | 8 | 168 | 176 | 1 | 4 | 47 | 52 |
| *Ochlerotatus argyrothorax* (Bonne-Wepster and Bonne) | 0 | 0 | 0 | 0 | 0 | 0 | 1 | 1 | 0 | 0 | 6 | 6 |
| *Orthopodomyia fascipes* (Coquillett) | 18 | 0 | 0 | 18 | 41 | 0 | 0 | 41 | 0 | 0 | 0 | 0 |
| *Sabethes albiprivus* Theobald | 14 | 5 | 0 | 19 | 9 | 0 | 3 | 12 | 0 | 1 | 0 | 1 |
| *Sa. amazonicus* Gordon and Evans | 1 | 0 | 0 | 1 | 0 | 0 | 0 | 0 | 0 | 0 | 0 | 0 |
| *Sa. belisarioi* Neiva | 0 | 0 | 0 | 0 | 0 | 0 | 0 | 0 | 3 | 0 | 0 | 3 |
| *Sa chloropterus* Humboldt | 3 | 0 | 1 | 4 | 8 | 2 | 1 | 11 | 7 | 4 | 0 | 11 |
| *Sa. cyaneus* (Fabricius) | 0 | 0 | 0 | 0 | 0 | 4 | 0 | 4 | 0 | 0 | 0 | 0 |
| *Sa. glaucodaemon* (Dyar and Shannon) | 2 | 0 | 0 | 2 | 5 | 4 | 0 | 9 | 4 | 3 | 0 | 7 |
| *Sa. tridentatus* Cerqueira | 55 | 7 | 13 | 75 | 53 | 8 | 2 | 63 | 68 | 82 | 21 | 171 |
| *Toxorhynchites ha. haemorrhoidalis* (Fabricius) | 2 | 15 | 89 | 106 | 2 | 7 | 99 | 108 | 7 | 17 | 105 | 129 |
| *Trichoprosopon digitatum* (Rondani) | 648 | 0 | 522 | 1,170 | 492 | 7 | 186 | 685 | 440 | 149 | 302 | 891 |
| *Wyeomyia aporonoma* Dyar and Knab | 5 | 9 | 6 | 20 | 4 | 2 | 1 | 7 | 7 | 9 | 0 | 16 |
| Total | 1,015 | 327 | 2,887 | 4,229 | 699 | 291 | 1,836 | 2,826 | 879 | 725 | 1,472 | 3,076 |

Distribution of diversity and abundance of mosquitoes collected during November 2017 and January-February 2018, found in the bamboo internode, plastic container, and tire traps, located in the forest, at the forest border, or in the peridomicile environment in the agrovillage of Rio Pardo, Presidente Figueiredo, Amazonas, Brazil.

*Traps

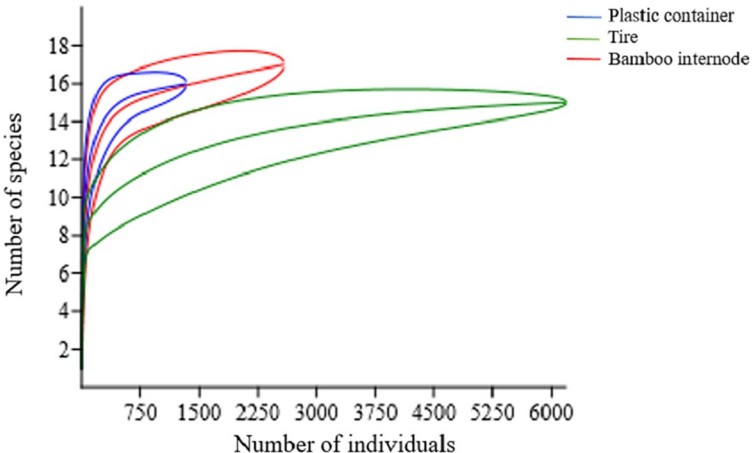

**Fig 3. Collected species of mosquitoes.** Rarefaction curves representing the accumulated richness of the immature mosquito species collected in plastic container, tire, and bamboo internode traps in the agrovillage of Rio Pardo, Presidente Figueiredo, Amazonas, Brazil periods of November 2017 and January–February 2018.

The species diversity of mosquitoes was significantly different between the bamboo internode, plastic container, and tire (PERMANOVA pseudo-F = 8.84090; p < 0.0005). At least 10 bioindicator species were identified according to the IndVal analysis. *Sabethes tridentatus*, *Tr. digitatum*, *Sa. albiprivus*, and *Or. fascipes* were the species that presented the greatest specificity and fidelity for bamboo internode larvitraps, while *Tx. ha. haemorrhoidalis*, *Cx. urichii*, *Li. flavisetosus*, *Li. durhamii*, *Cx. nigripalpus*, and *Ae. albopictus* had the greatest specificity and fidelity for the tire trap. There were no bioindicator species for the plastic container larvitrap (Table 3).

Forest, forest edge, and peridomicile environments presented differences in the diversity of immature mosquitoes (PERMANOVA pseudo-F = 3.22809; p < 0.0010). The forest edge environment had the greatest diversity of species, followed by the peridomicile and forest (Table 4). In all environments, tire larvitraps collected the largest number of individuals, while at the forest edge, the plastic container and tire larvitraps collected the greatest number of species (Table 1).

The diversity of species among larvitraps installed in different environments, were presented in the following order: forest: plastic container > tire > bamboo internode; forest edge: tire > plastic container > bamboo internode; peridomicile: plastic container > tire > bamboo internode (Table 4, Fig 4).

The equitability of species in the forest environment was highest in plastic container larvitraps and lowest in bamboo internode larvitraps. At the forest edge, the highest equitability value was observed in tires and the lowest in bamboo internode larvitraps. In the peridomicile, equitability was higher in tire larvitraps and lower in bamboo internode larvitraps (Table 4).

**Table 2. Diversity analyses of immature mosquitoes.**

| Ecology attributes | Bamboo internode trap | Plastic container trap | Tire trap |
|---|---|---|---|
| Shannon—Wienner | 1.527 | 2.07 | 1.741 |
| Simpson | 0.6075 | 0.8268 | 0.7439 |
| Equitability | 0.5389 | 0.7467 | 0.6429 |

Ecological indices of species collected during November 2017 and January-February 2018, found in larvitraps installed in the agrovillage of Rio Pardo, Presidente Figueiredo, Amazonas, Brazil.

**Table 3. Indication value of species (IndVal) of immature mosquitoes.**

| Specie | Larvitraps | IndVal% | P | Frequency |
|---|---|---|---|---|
| *Sabethes tridentatus* Cerqueira | Bamboo internode | 38.0 | 0.027 | 39 |
| *Trichoprosopon digitatum* (Rondani) | Bamboo internode | 33.5 | 0.044 | 33 |
| *Sa. albiprivus* Theobald | Bamboo internode | 27.0 | 0.012 | 12 |
| *Orthopodomyia fascipes* (Coquillett) | Bamboo internode | 16.7 | 0.036 | 4 |
| *Toxorhynchites ha. haemorrhoidalis* (Fabricius) | Tire | 85.5 | 0.001 | 49 |
| *Culex urichii* (Coquillett) | Tire | 74.3 | 0.001 | 37 |
| *Limatus flavisetosus* Oliveira Castro | Tire | 53.3 | 0.001 | 25 |
| *Li. durhamii* Theobald | Tire | 52.0 | 0.001 | 46 |
| *Cx. nigripalpus* Theobald | Tire | 43.3 | 0.005 | 35 |
| *Aedes albopictus* (Skuse) | Tire | 37.0 | 0.010 | 26 |

Bioindicator species of bamboo internode and tire larvitraps, according to IndVal, collected during November 2017 and January-February 2018, in the agrovillage of Rio Pardo, Presidente Figueiredo, Amazonas, Brazil.

## Discussion

The species of mosquitoes collected in this study represented 7% of the mosquito fauna found in the Amazonas state, which is approximately 270 species [39–48]. Rarefaction curves indicated the stabilization of the species diversity to 20 species collected in the agrovillage, during the period of study, by the collection method used. However, Soares [49] and Pereira-Silva [12] found a higher diversity (40 and 46 species, respectively), in the same study area but using CDC light traps and human landing protected methods. Also, Abad-Franch et al. [11] registered 13 genera of mosquitoes but they were not identified to species level.

The high abundance of *Cx. urichii* has not been recorded in previous studies of mosquito ecology in the Brazilian Amazon [50, 51]. However, at low frequencies, this species can be observed in natural and artificial breeding sites in forest areas of the Manaus municipality, Amazonas [52, 53]. *Trichoprosopon digitatum*, followed by *Cx.* (*Melanoconion*) spp., were the second and third species with the largest number of specimens. Chaverri et al. [54] also reported the dominance of *Tr. digitatum*, collected in ovitraps in a forest region in Costa Rica. Due to the difficulties of taxonomic characterization of *Cx.* (*Melanoconion*) species, these were only identified at the section level. However, a high abundance of subgenus species was reported in the studies of Hutchings et al. [43, 55] carried out along Amazonian rivers, as well as in the work of Ribeira et al. [56] in areas of the Atlantic Forest of São Paulo.

Larvitrap colonization varied depending on the preference of the species. A greater abundance was recorded in tire traps, while the highest diversity was observed in plastic container

**Table 4. Diversity analyses of immature mosquitoes.**

| Ecological attributes | Forest | | | | Forest edge | | | | Peridomicile | | | |
|---|---|---|---|---|---|---|---|---|---|---|---|---|
| | Bamboo internode* | Plastic container* | Tire* | Environment Forest | Bamboo internode | Plastic container | Tire | Environment Forest edge | Bamboo internode | Plastic container | Tire | Environment Peridomicile |
| Shannon—Wienner | 1.24 | 1.41 | 1.35 | 1.64 | 1.19 | 1.75 | 1.86 | 2.04 | 1.71 | 2.01 | 1.88 | 2.03 |
| Simpson | 0.55 | 0.59 | 0.59 | 0.72 | 0.48 | 0.73 | 0.8 | 0.83 | 0.70 | 0.83 | 0.81 | 0.83 |
| Equitability | 0.46 | 0.59 | 0.56 | 0.59 | 0.46 | 0.64 | 0.68 | 0.7 | 0.64 | 0.76 | 0.81 | 0.73 |

Ecological indices of species collected during November 2017 and January-February 2018, found in larvitraps installed in forest, forest edge, and peridomicile environments in the agrovillage of Rio Pardo, Presidente Figueiredo, Amazonas, Brazil.

* Traps.

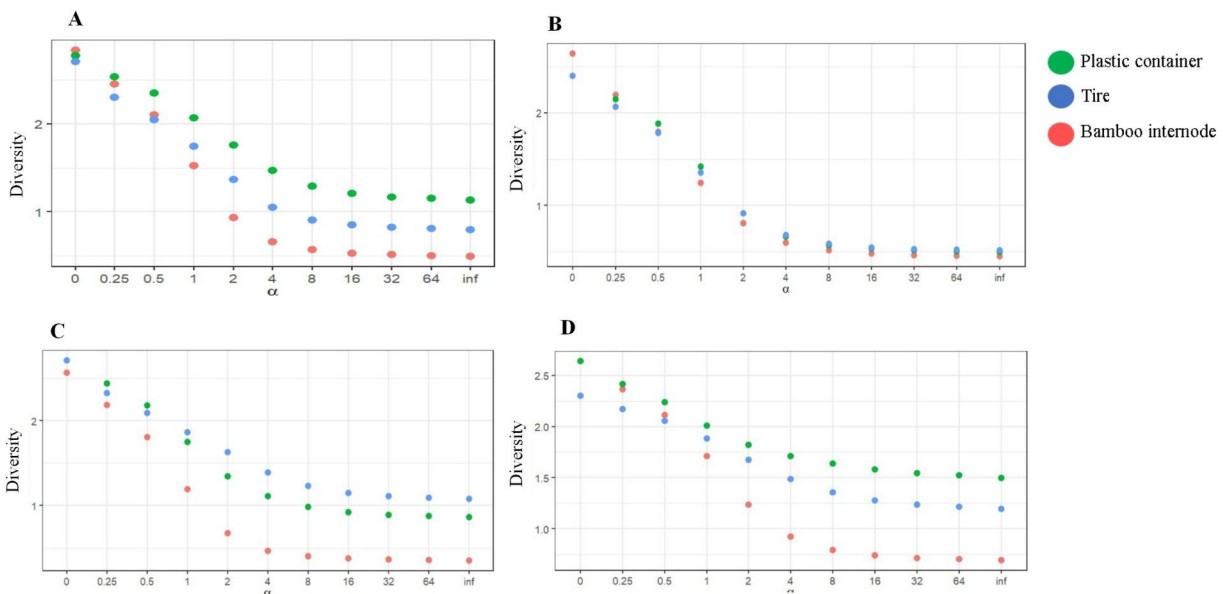

**Fig 4. Rényi diversity profiles of immature mosquitoes. A**—Diversity of species found in bamboo internodes, plastic containers, and tire larvitraps. **B**—Diversity of species found in bamboo internodes, plastic containers, and tire larvitraps installed in the forest environment. **C**—Diversity of species found in bamboo internodes, plastic containers, and tire larvitraps installed in the forest edge environment. **D**—Diversity of species found in bamboo internodes, plastic containers, and tire larvitraps installed in the peridomicile environment. X axis: Alpha (α) zero = log wealth, 1 = Shannon Index, 2 = Simpson Index, Inf = Berger Parker Index. Rio Pardo, Presidente Figueiredo, Amazonas.

traps. Similar results of high mosquito abundance in tires were reported by Lopes et al. [15, 23]. However, these studies also described the artificial tire trap as hosting the greatest diversity of mosquito species when compared to plastic containers, bamboo, and aluminum cans. Lopes et al. [15] also found that the similarity of species was greatest between the tire and bamboo traps, inconsistent with the current results, which indicate that the greatest similarity of immature mosquito species was observed between plastic container and tire larvitraps.

The composition of immature mosquito species was different between larvitrap types, as was also observed in investigations of culicid fauna carried out by Calado and Silva [57] and Lopes [23]. Some of the species, as revealed by IndVal, were frequent in breeding sites following patterns commonly described in the literature, as in the case of *Sa. tridentatus*, *Tr. digitatum*, *Sa. albiprivus*, and *Or. fascipes*, generally found in natural breeding sites such as the bamboo internode and *Li. durhamii*, *Cx. nigripalpus*, and *Ae. albopictus*, observed in artificial breeding sites, such as tires [27, 31, 57–59]. *Toxorhynchites ha. haemorrhoidalis*, *Cx. urichii*, and *Li. flavisetosus* have not yet been associated with tires as their breeding site.

The greatest diversity of immature mosquitoes species found at the edge of the forest suggested a change in the habitat of these populations, and similar results were reported in the study of Steiger et al. [60]. Chaverri et al. [54], in their study of immature mosquito fauna in different environments, observed no difference in mosquito populations between the primary and secondary forest environment. Ribeiro et al. [56] suggested that environmental stresses increase the number of niches favorable to mosquitoes and, thus, promote the greater diversity of mosquitoes in anthropogenic environments.

Tire larvitraps, when compared to other types present in the forest, forest edge, and peridomicile environments, had the highest species abundance in all environments, as well as the greatest wealth of species in the forest edge environment. It is believed that species that develop in tires originally breed in holes of trees and have a preference for tires due to their similar characteristics, such as a dark environment, shading, presence of organic matter retained

inside, retention of excessive water volume, and a slow evaporation process [23, 61, 62]. According to Beier et al. [62], some species ended up dominating the tire trap, and, in the case of this study, the dominant species was *Cx. urichii*. Rubio et al. [63], evaluated the success of tire colonization across a gradient of less and more urbanized areas in Argentina, and they found that the majority of species captured frequented both areas. In addition, there was a trend of increasing abundance of vector species in less urbanized environments.

In the forest edge environment, culicid species found in plastic container larvitraps had a species richness similar to that of mosquito populations in tire traps. However, the fact that there was a low abundance of 10 species present in the forest edge tire larvitraps, with values between one and nine individuals per species, may suggest that these species preferred the plastic container in this environment for opportunistic colonization, since the forest edge did not provide a large variety of natural breeding sites as the forest would.

In general, in all the studied environments, artificial larvitraps, namely the plastic containers and tires, had higher immature mosquito species diversity and equitability values when compared to bamboo internode larvitraps.

Although there is a great diversity of breeding sites and different patterns of choice for oviposition site which vary between mosquito species, artificial larvitraps presented the best conditions for the development of immature mosquitoes, independently of the environment. Thus, it is plausible to suggest that some of the species collected in the current study demonstrated a certain eclecticism at the time of choosing the place of oviposition and had success in the development of their larvae in artificial breeding sites. In addition, the smaller diversity of species found in the bamboo internode larvitraps in all environments reinforces the idea that only a few species, especially those considered wilder, such as the Sabethini tribe, have a preference for small natural breeding sites found in wild environments [27, 64].

All vector species found in this study were collected in plastic containers and tire larvitraps, with a greater abundance in the forest edge and peridomicile environments. Only six of the twenty species collected here have been registered as vectors of different pathogenic agents to humans. *Haemagogus janthinomys*, *Sa. chloropterus* and *Sa. cyaneus* are vectors in the wild transmission cycle of Yellow Fever, an endemic disease in the Amazon region which periodically causes outbreaks or epidemics [65]. In recent years, cases of the disease have been recorded in regions considered non-endemic in Brazil, such as São Paulo, Rio Grande do Sul and Rio de Janeiro. Currently 1,281 cases and 14 deaths have been reported in the country, with the highest number of cases in the South and Southeast [66–68]. *Culex nigripalpus* and *Cx. quinquefasciatus* carry the São Luis Encephalitis virus through a zoonotic cycle, and it is widely distributed in the Americas, with sporadic outbreaks occurring in Brazil since 1960, especially in the Brazilian Amazon region [69, 70]. Cases of São Luis Encephalitis in humans in Brazil are scarce, however they have been recorded in patients in Pará, São Paulo and in a dengue epidemic in the municipality of São José do Rio Preto, state of São Paulo [71–73]. *Aedes albopictus* is a secondary vector of the Dengue virus. Both are found in several regions of the world and cause serious public health problems. In Brazil, 823,738 probable cases were reported, with an incidence rate of 392.0 cases per 100 thousand inhabitants and the most affected regions were the Midwest (997.6 cases/100 thousand inhabitants) and the South (897.5 cases/100 thousand population) [74].

All vector species found in this study were collected in plastic containers and tire larvitraps, with a greater abundance in the forest edge and peridomicile environments. These observations suggest that such artificial breeding sites may be risk factors for infection with typical wild arboviruses [58, 75].

Lopes [22] actively searched for immature mosquitoes in artificial breeding sites in rural areas of Londrina, Paraná, and showed that the abundance of immature mosquitoes was

higher in containers of water, followed by tires and troughs. In addition, species *Cx. quinque-fasciatus* and *Li. durhamii* were found in all artificial breeding sites, and, thus, Lopes and colleagues suggested that these were species adapted to anthropogenic environments.

Through this study it was possible to find some differences in the diversity of mosquito species that colonize larvitraps located in environments with different degrees of anthropization. The preference for artificial larvitraps observed in the current study may be associated with the opportunism of females or may indicate a change in the habits of these wild species, including vector species (e.g., the finding of larvae of *Hg. janthinomys*, an acrodendrophilic species that uses phytotelma or the hollow of a tree as a breeding site), in larvitraps in the peridomiciliary area. However, further studies are required to assess these behaviors.

## Supporting information

**S1 Table. Data from mosquito collections, carried out in Rio Pardo, Presidente Figueiredo, Amazonas, Brazil.**
(XLSX)

## Acknowledgments

We thank Fernanda Fonseca MSc for providing data on the territorial delimitation of the Rio Pardo settlement, Dr. Bernardo Horta and Antônio Balieiro MSc for their assistance in the statistical analysis, Eric Marialva MSc, Antônio Leão MSc, Ricardo Mota, and Sebastião Dias for helping with field collections, and Gervilane Ribeiro for mosquito class identification. In memory of Patrícia Dantas, we would like to thank her for helping to make the traps and growing the immature mosquitoes in the laboratory. We would like to thank anonymous referees for their comments and Editage (www.editage.com) for English language editing.

## Author Contributions

**Conceptualization:** Jessica Feijó Almeida, Claudia María Ríos-Velásquez, Felipe Arley Costa Pessoa.

**Data curation:** Jessica Feijó Almeida, Felipe Arley Costa Pessoa.

**Formal analysis:** Jessica Feijó Almeida.

**Funding acquisition:** Jessica Feijó Almeida, Felipe Arley Costa Pessoa.

**Methodology:** Jessica Feijó Almeida, Heliana Christy Matos Belchior, Claudia María Ríos-Velásquez, Felipe Arley Costa Pessoa.

**Project administration:** Jessica Feijó Almeida, Felipe Arley Costa Pessoa.

**Writing – original draft:** Jessica Feijó Almeida, Felipe Arley Costa Pessoa.

**Writing – review & editing:** Jessica Feijó Almeida, Heliana Christy Matos Belchior, Claudia María Ríos-Velásquez, Felipe Arley Costa Pessoa.

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
