## [Decision Letter · Decision Letter 0]

20 Jul 2020

PONE-D-20-18942

Diversity of mosquitoes (Diptera: Culicidae) collected in different types of larvitraps in an Amazon rural settlement

PLOS ONE

Dear Dr. Almeida,

Thank you for submitting your manuscript to PLOS ONE. After careful consideration, we feel that it has merit but does not fully meet PLOS ONE’s publication criteria as it currently stands. Therefore, we invite you to submit a revised version of the manuscript that addresses the points raised during the review process.

Both reviewers have recommended more details on the methodology for a better and fair evaluation of manuscript results. Besides, I suggest this manuscript goes through a native English speaker, preferentially someone with a desirable background in mosquito ecology before resubmission. 

We look forward to receiving your revised manuscript.

Kind regards,

Rafael Maciel-de-Freitas

Academic Editor

PLOS ONE

Journal Requirements:

2. In your Methods section, please provide additional location information of the study site, including geographic coordinates for the data set if available.

"We thank for the grant of the scientific initiation scholarship of the co-author of the article, Heliana

Belchior nº 00812016 - PAIC - AM – 2016. We would also like to thank Fiocruz / VPEIC

for the scholarship used to carry out the study and the post-graduate course in Condições

de Vida e Situações de Saúde na Amazônia of the Instituto Lebônidas e Maria Deane -

Fiocruz Amazônia, for the training and qualification of Master in Public Health."

"This work is funded by the Fundanção de Amparo a Pesquisa do Estado do Amazonas – FAPEAM through the Programa de Excelência em Pesquisa Básica e Aplicada em Sáude (public call nº 004/2014). The article is a by-product of the umbrella project "Arboviroses emerging in the Amazon: Alphavirus incidence risk factors with an emphasis on Mayaro, on the agricultural frontier in Central Amazon ", under the coordination of FACP. Information for this grant can be found at http://www.fapeam.am.gov.br/wp-content/uploads/2014/07/DecCD-159_2014.pdf. The funders had no role in study design, data collection and analysis, decision to publish, or preparation of the manuscript."

4. We note that Figure 1 in your submission contain satellite images which may be copyrighted. All PLOS content is published under the Creative Commons Attribution License (CC BY 4.0), which means that the manuscript, images, and Supporting Information files will be freely available online, and any third party is permitted to access, download, copy, distribute, and use these materials in any way, even commercially, with proper attribution. For these reasons, we cannot publish previously copyrighted maps or satellite images created using proprietary data, such as Google software (Google Maps, Street View, and Earth). For more information, see our copyright guidelines: http://journals.plos.org/plosone/s/licenses-and-copyright.

4.1.    You may seek permission from the original copyright holder of Figure 1 to publish the content specifically under the CC BY 4.0 license.

4.2.    If you are unable to obtain permission from the original copyright holder to publish these figures under the CC BY 4.0 license or if the copyright holder’s requirements are incompatible with the CC BY 4.0 license, please either i) remove the figure or ii) supply a replacement figure that complies with the CC BY 4.0 license. Please check copyright information on all replacement figures and update the figure caption with source information. If applicable, please specify in the figure caption text when a figure is similar but not identical to the original image and is therefore for illustrative purposes only.

5. We note that Figure 2 includes an image of a participant in the study. 

Reviewers' comments:

Reviewer's Responses to Questions

**Comments to the Author**

1. Is the manuscript technically sound, and do the data support the conclusions?

Reviewer #1: Partly

Reviewer #2: Partly

2. Has the statistical analysis been performed appropriately and rigorously? 

Reviewer #1: Yes

Reviewer #2: Yes

3. Have the authors made all data underlying the findings in their manuscript fully available?

Reviewer #1: Yes

Reviewer #2: No

4. Is the manuscript presented in an intelligible fashion and written in standard English?

Reviewer #1: Yes

Reviewer #2: No

5. Review Comments to the Author

Reviewer #1: The MS “Diversity of mosquitoes (Diptera: Culicidae) collected in different types of larvitraps in an Amazon rural settlement” is relevant and agrees with the purposes of the Journal. However, I recommend 'Major Revision' modifications, as follows:

Comments

Line 33. Change "Trichoprosopom digitatum" to "Trichoprosopon digitatum". Melanoconion in “Cx. (Melanoconion) sp.”, written in italics, please.

Lines 92 and 93. Do you mean "by" here? Change "Koppen" to "Köppen” climate classification".

Lines 101-102. Include in the text a reference about these deforestation data in the sampling area.

Lines 103-106. Authors need to include the geographic coordinates of the collection points and characterize the different collection points as well as information on the vegetation cover of the sampling area.

Line 109. Sampling was carried out for a short period. I do not consider that it allowed a conclusion on distribution patterns and knowledge of species diversity.

Line 110. What is the size of the container used? Were they plastic containers?

Lines 110-11. Were the traps (bamboo internodes, tires, and plastic containers) reused? If the answer is positive, please make it clear how the treatment was performed for relocation at the sampling points?

Line 120. Specify the type of plastic transport bag for the immature.

Line 123. How were the larvae fed?

Line 126. Incorrect reference citations, actually, there are two volumes:

Lane, J. 1953a. Neotropical Culicidae - 1st Volume. University of São Paulo, São Paulo. (and)

Lane J. 1953b. Neotropical Culicidae - 2nd Volume. University of São Paulo, São Paulo.

Line 128. In which entomological collection were the specimens collected deposited?

Lines 126-128. Authors must include the following bibliographic reference: J F Reinert. J Am Mosq Control Assoc. 2000 Sep;16(3):175-88. New Classification for the Composite Genus Aedes (Diptera: Culicidae: Aedini), Elevation of Subgenus Ochlerotatus to Generic Rank, Reclassification of the Other Subgenera, and Notes on Certain Subgenera and Species

Line 147. Melanoconion in “Cx. (Melanoconion) sp.” written in italics, please.

Lines 154-156. Why did you use "and" in the text to separate names of the authors of species and here, in the table you used "&"? You must see the instructions of the Journal, and uniform it.

Table 1-4. Inform the sampling period.

Table 1. Sa. chloropterus Von Humboldt. It is double wrong: firstly, it should be written "von" with the initial letter in lower case; secondly, it is a German a preposition between the name and the surname (as "de" in Portuguese in names such as "João de Almeida"); therefore, the citation of the surname MUST NOT BEGIN with the preposition, just with the surname: Humboldt.

Table 1. Toxorhynchites haemorrhoidalis haemorrhoidalis (Fabricius). In order to make it more concise, in the table, since it is the nominotypical subspecies, you could abbreviate this specific name, which is the same as the subespecific.

Line 185. Tx. haemorrhoidalis haemorrhoidalis. In order to make it more concise, since it is the nominotypical subspecies, you could abbreviate this specific name, which is the same as the subespecific.

Line 232. Cx. (Melanoconion) spp.: Melanoconion in italics; it is a subgeneric name!!!!!

Line 235. Did the authors obtain male specimens to identify at a specific level?

Line 255. Toxorhynchites haemorrhoidalis haemorrhoidalis, why not abbreviate the name of this genus if you are doing it for all other genus?

Discussion: The authors poorly discussed the medical importance of the mosquito species found in this study.

Lines 306-310. The conclusion was not clear, the sampling was carried out in a short period of collection. Therefore, how is it possible to associate a change in the oviposition behavior of some vector species of etiologic agents?

Line 391. Aedes (Stegomyia) aegypti: Stegomyia in italics; it is a subgeneric name!!!!!

Line 411. Change "Neotropical Culioidae" to "Neotropical Culicidae".

Reviewer #2: In general, the manuscript is sound and has new biological information. There are some weak parts that demand more work and time from authors. I will mention a list of points that they need to consider before the MS is accepted for publication.

1- Introduction - "Rural settlements are established in places that are favorable for human-vector interaction due to their proximity to extensive forest areas and the lack of adequate infrastructure for the residents who live there." I partially disagree with authors about their argument. The places are favorable or become favorable because of anthropogenic changes in the forest environments?

2- What do authors mean by "the habitats of residents? Why habitat? Can it be socioecological conditions?

3- Authors need to adjust the use of some words. For instance, it is not common to say "Mosquitoes are characterized by their flexible use of a wide variety of breeding sites," The word flexible is not adequate, there are better word to use. i.e. generalist. Breeding sites can be replaced by larval habitats, adult habitat.

4- Please consider a better way to mention "from larges found at the soil level", ground pools, standing water, others? Authors must check the terminology used in mosquito ecology.

5- Authors must consider to use the correct ecological term for "small pools of water stored in leaves in the canopy of trees." Can it be phytotelma?

6- Immature forms of Anopheles species have been recorded in artificial containers [16,17], including An. darlingi Root, which was found in an artificial lagoon in the urban region of Manaus in the state of Amazonas. Please rewrite this part, because an artificial lagoon is not an artificial container.

7- In the last paragraph of the introduction, authors say "Due to the importance of sanitary conditions and the ecological evolution that has led to the use of water containers for breeding by female mosquitoes, this work aimed to identify mosquito species that colonize different types of artificial breeding places, located in environments with different levels of anthropization within the state of Amazonas."

This entire paragraph needs to be rewritten. What do authors mean when they say sanitary conditions and ecological evolution? What is a water container? Why breeding by females? Can female breed on water containers? Can you replace artificial breeding places by larval habitats? Because your investigation is limited to a single settlement in Presidente Figueiredo, I consider it is incorrect to say, "within the state of Amazonas". Your sample is small, and it is does not represent the immense diversity found in the Amazon. Please considering the focus of your study, rewrite this part.

Materials and Methods

1- Can you please provide better details of the climate in the region studied?

2- Please clarify your procedure to choose locations to install your traps. Did you used satellite images to select locations within the settlements area? How did you select locations within the settlement? You said you installed 9 traps per sampling point, please define what you mean by sampling point. Please, add this information, they are essential for an ecological study.

3- Considering you used tap water to fill the traps, do you think your sampling was biased based on the female choice that depends on several molecules present in the habitats, including pheromones. How the water characteristics may have affect your sampling diversity and abundance and occurrence of mosquitoes?

4- What do you mean by material collection? Please clarify.

5- How did you define the 15 days interval between installation and collection days?

6- I assume you installed 162 traps each field collection. Did you install all traps in a single day?

7- How did you avoided cross contamination among traps?

8- A believe the larvae were in early stages and most have died during the trip and in the lab. Have you counted and identified all of them? This needs to be clarified because of your statistical analyses and diversity indexes you used.

9- Figure 3 legend needs to be together with the graphic.

Discussion

1- It is of paramount importance to explain why authors consider that the larvitraps were good to sample the most common mosquito species in the Rio Pardo agrovillage. Can you please provide reference (in addition to the thesis you cited) of other published studies on the same agrovillage? How can you support your argument that traps recovered the most common species? Please add reference. How can you be sure your trap collections in artificial habitats were able to return most species present in the region?

6. PLOS authors have the option to publish the peer review history of their article (what does this mean?). If published, this will include your full peer review and any attached files.

Reviewer #1: No

Reviewer #2: No

---

## [Author Response · Author response to Decision Letter 0]

15 Sep 2020

Editor

Please ensure that your manuscript meets PLOS ONE's style requirements, including those for file naming. The PLOS ONE style templates can be found at https://journals.plos.org/plosone/s/file?id=wjVg/PLOSOne_formatting_sample_main_body.pdf andhttps://journals.plos.org/plosone/s/file?id=ba62/PLOSOne_formatting_sample_title_authors_affiliations.pdf

Ok, adjusted

Editor: In your Methods section, please provide additional location information of the study site, including geographic coordinates for the data set if available. Answer: Ok, adjusted Thank you for stating the following in the Acknowledgments Section of your manuscript: "We thank for the grant of the scientific initiation scholarship of the co-author of the article, Heliana Belchior nº 00812016 - PAIC - AM – 2016. We would also like to thank Fiocruz / VPEIC for the scholarship used to carry out the study and the post-graduate course in Condições de Vida e Situações de Saúde na Amazônia of the Instituto Lebônidas e Maria Deane - Fiocruz Amazônia, for the training and qualification of Master in Public Health." We note that you have provided funding information that is not currently declared in your Funding Statement. However, funding information should not appear in the Acknowledgments section or other areas of your manuscript. We will only publish

funding information present in the Funding Statement section of the online submission form. Please remove any funding-related text from the manuscript and let us know how you would like to update your Funding Statement. Currently, your Funding Statement reads as follows: "This work is funded by the Fundanção de Amparo a Pesquisa do Estado do Amazonas – FAPEAM through the Programa de Excelência em Pesquisa Básica e Aplicada em Sáude (public call nº 004/2014). The article is a by-product of the umbrella project "Arboviroses emerging in the Amazon: Alphavirus incidence risk factors with an emphasis on Mayaro, on the agricultural frontier in Central Amazon ", under the coordination of FACP. Information for this grant can be found at http://www.fapeam.am.gov.br/wp-content/uploads/2014/07/DecCD-159_2014.pdf. The funders had no role in study design, data collection and analysis, decision to publish, or preparation of the manuscript." Answer: Ok, adjusted. Financing information will be added to funding Statement section of the online submission form We note that Figure 1 in your submission contain satellite images which may be copyrighted. All PLOS content is published under the Creative Commons Attribution License (CC BY 4.0), which means that the manuscript, images, and Supporting Information files will be freely available online, and any third party is permitted to access, download, copy, distribute, and use these materials in any way, even commercially, with proper attribution. For these reasons, we cannot publish previously copyrighted maps or satellite images created using proprietary data, such as Google software (Google Maps,

Street View, and Earth). For more information, see our copyright guidelines: http://journals.plos.org/plosone/s/licenses-and-copyright. We require you to either (1) present written permission from the copyright holder to publish these figures specifically under the CC BY 4.0 license, or (2) remove the figures from your submission: Answer: Ok, adjusted We note that Figure 2 includes an image of a participant in the study. As per the PLOS ONE policy (http://journals.plos.org/plosone/s/submission-guidelines#loc-human-subjects-research) on papers that include identifying, or potentially identifying, information, the individual(s) or parent(s)/guardian(s) must be informed of the terms of the PLOS open-access (CC-BY) license and provide specific permission for publication of these details under the terms of this license. Please download the Consent Form for Publication in a PLOS Journal (http://journals.plos.org/plosone/s/file?id=8ce6/plos-consent-form-english.pdf). The signed consent form should not be submitted with the manuscript, but should be securely filed in the individual's case notes. Please amend the methods section and ethics statement of the manuscript to explicitly state that the patient/participant has provided consent for publication: “The individual in this manuscript has given written informed consent (as outlined in PLOS consent form) to publish these case details”. If you are unable to obtain consent from the subject of the photograph, you will need to remove the figure and any other textual identifying information or case descriptions for this individual.

Answer: Ok, adjusted

Editor 2

(1) Thank you again for removing the following funding information from the Acknowledgments section of your manuscript:

“We thank for the grant of the scientific initiation scholarship of the co-author of the article, Heliana Belchior nº 00812016 - PAIC - AM – 2016. We would also like to thank Fiocruz / VPEIC for the scholarship used to carry out the study and the post-graduate course in Condições de Vida e Situações de Saúde na Amazônia of the Instituto Lebônidas e Maria Deane - Fiocruz Amazônia, for the training and qualification of Master in Public Health.”

-However, we note your Financial Disclosure statement in the online submission form still reads as follows and does not include the funding information removed from your Acknowledgments:

“This work is funded by the Fundanção de Amparo a Pesquisa do Estado do Amazonas – FAPEAM through the Programa de Excelência em Pesquisa Básica e Aplicada em Sáude (public call nº 004/2014). The article is a by-product of the umbrella project "Arboviroses emerging in the Amazon: Alphavirus incidence risk factors with an emphasis on Mayaro, on the agricultural frontier in Central Amazon ", under the coordination of FACP. Information for this grant can be found at http://www.fapeam.am.gov.br/wp-content/uploads/2014/07/DecCD-159_2014.pdf. The funders had no role in study design, data collection and analysis, decision to publish, or preparation of the manuscript.”

-Before we proceed, we’ll require some additional information regarding the funding received for your study.

1.) Please confirm whether you received the following funding for this study: the scientific initiation scholarship of the co-author of the article, Heliana Belchior nº 00812016 - PAIC - AM – 2016.

2.) If you received the following funding for this study, please provide the full name of the organization that provided the scholarship: the scientific initiation scholarship of the co-author of the article, Heliana Belchior nº 00812016 - PAIC - AM – 2016.

3.) Please confirm whether you received the following funding for this study: Fiocruz / VPEIC for the scholarship used to carry out the study and the post-graduate course in Condições de Vida e Situações de Saúde na Amazônia of the Instituto Lebônidas e Maria Deane - Fiocruz Amazônia, for the training and qualification of Master in Public Health.

4.) If you received the following funding for this study, please provide the full name of the organization that provided the scholarship: Fiocruz / VPEIC

for the scholarship used to carry out the study and the post-graduate course in Condições de Vida e Situações de Saúde na Amazônia of the Instituto Lebônidas e Maria Deane - Fiocruz Amazônia, for the training and qualification of Master in Public Health.

5.) If you received the following funding for this study, please provide the name of the author that received the scholarship and the associated scholarship number: Fiocruz / VPEIC for the scholarship used to carry out the study and the post-graduate course in Condições de Vida e Situações de Saúde na Amazônia of the Instituto Lebônidas e Maria Deane - Fiocruz Amazônia, for the training and qualification of Master in Public Health.

6.) Please provide the name of the author that received the following funding: Fundanção de Amparo a Pesquisa do Estado do Amazonas – FAPEAM through the Programa de Excelência em Pesquisa Básica e Aplicada em Sáude (public call nº 004/2014).

Answer: The requested information was added to the system, as noted in the print, however it is not being generated in the final pdf.

(2) Thank you for stating the following in the Methods section:

"The study was carried out in the Rio Pardo rural settlement, Presidente Figueiredo Municipality, State of Amazonas, Brazil (01º49’02.4” S, 060º19’03.6” W) and borders the Canoas settlement, the Waimiri-Atroari indigenous reserve and private land, and the Brazilian Federal Government (Fig1)."

"The collection was made with the permanent license, nº 12186, issued by SISBIO-ICMBio. The individual in this manuscript has given written informed consent (as outlined in PLOS consent form) to publish these case details (Fig. 2)."

However, we note the following is included in your submission Ethics statement:

"The permanent mosquito collection collection license belongs to FACP, nº 12186, issued by SISBIO-ICMBio.

The project is authorized by the Research Ethics Committee Nº CAAE 40108114.7.0000.5248 and permission issued by the community president, Mr. Eleonildis Fernandes da Silva".

Answer: Ok, adjusted 

Reviewer#1

Reviewer1. Line 33. Change "Trichoprosopom digitatum" to "Trichoprosopon

digitatum". Melanoconion in “Cx. (Melanoconion) sp.”, written in italics, please.

Answer: ok, adjusted

Reviewer1. Lines 92 and 93. Do you mean "by" here? Change "Koppen" to "Köppen”

climate classification".

Answer: Ok, adjusted

Reviewer1. Lines 101-102. Include in the text a reference about these deforestation data

in the sampling area.

Answer: ok, adjusted

Reviewer1. Lines 103-106. Authors need to include the geographic coordinates of the

collection points and characterize the different collection points as well as information on

the vegetation cover of the sampling area.

Answer: ok, adjusted

Reviewer1. Line 109. Sampling was carried out for a short period. I do not consider that

it allowed a conclusion on distribution patterns and knowledge of species diversity.

Answer: ok, adjusted

Reviewer1. Line 110. What is the size of the container used? Were they plastic

containers?

Answer: Three different types of traps were used; bamboo internode Guandua sp. (about 32 cm in height and 93 cm in diameter); rubber tire (35,6 cm, cut in half); plastic container (about 11 cm high and 10 cm in diameter)

Reviewer1. Lines 110-11. Were the traps (bamboo internodes, tires, and plastic containers) reused? If the answer is positive, please make it clear how the treatment was performed for relocation at the sampling points?

Answer: ok, adjusted

Reviewer1. Line 120. Specify the type of plastic transport bag for the immature.

Answer: ok, adjusted

Reviewer1. Line 123. How were the larvae fed?

Answer: ok, adjusted Reviewer1: Line 126. Incorrect reference citations, actually, there are two volumes: Lane, J. 1953a. Neotropical Culicidae - 1st Volume. University of São Paulo, São Paulo. (and) Lane J. 1953b. Neotropical Culicidae - 2nd Volume. University of São Paulo, São Paulo. Answer: Ok, adjusted Reviewer1: Line 128. In which entomological collection were the specimens collected deposited? Answer: Deposited in the collection of the Instituto Leônidas e Maria Deane/Laboratório de Ecologia e Doenças Transmissíveis da Amazônia (ILMD - LEDTA), Fiocruz Amazônia. Adjusted.

Reviewer1: Lines 126-128. Authors must include the following bibliographic reference: J F Reinert. J Am Mosq Control Assoc. 2000 Sep;16(3):175-88. New Classification for the Composite Genus Aedes (Diptera: Culicidae: Aedini), Elevation of Subgenus Ochlerotatus to Generic Rank, Reclassification of the Other Subgenera, and Notes on Certain Subgenera and Species.

Answer: ok, adjusted

Reviewer1. Line 147. Melanoconion in “Cx. (Melanoconion) sp.” written in italics, please.

Answer: ok, adjusted

Reviewer1. Lines 154-156. Why did you use "and" in the text to separate names of the authors of species and here, in the table you used "&"? You must see the instructions of the Journal, and uniform it.

Answer: ok, adjusted

Reviewer1: Table 1-4. Inform the sampling period.

Answer: ok, adjusted

Reviewer1: Table 1. Sa. chloropterus Von Humboldt. It is double wrong: firstly, it should be written "von" with the initial letter in lower case; secondly, it is a German a preposition between the name and the surname (as "de" in Portuguese in names such as "João de Almeida"); therefore, the citation of the surname MUST NOT BEGIN with the preposition, just with the surname: Humboldt.

Answer: ok, adjusted

Reviewer1: Table 1. Toxorhynchites haemorrhoidalis haemorrhoidalis (Fabricius). In order to make it more concise, in the table, since it is the nominotypical subspecies, you could abbreviate this specific name, which is the same as the subespecific.

Answer: ok, adjusted

Reviewer1: Line 185. Tx. haemorrhoidalis haemorrhoidalis. In order to make it more concise, since it is the nominotypical subspecies, you could abbreviate this specific name, which is the same as the subespecific.

Answer: ok, adjusted

Reviewer1: Line 232. Cx. (Melanoconion) spp.: Melanoconion in italics; it is a subgeneric name!!!!!

Answer: ok, adjusted

Reviewer1: Line 235. Did the authors obtain male specimens to identify at a specific level?

Answer: yes, but we did not get to identify the adult species of this subgenus

Reviewer1: Line 255. Toxorhynchites haemorrhoidalis haemorrhoidalis, why not abbreviate the name of this genus if you are doing it for all other genus?

Answer: ok, adjusted

Reviewer1: Discussion: The authors poorly discussed the medical importance of the mosquito species found in this study.

Answer: ok, adjusted

Reviewer1: Lines 306-310. The conclusion was not clear, the sampling was carried out in a short period of collection. Therefore, how is it possible to associate a change in the oviposition behavior of some vector species of etiologic agents?

Answer: ok, adjusted

Reviewer1: Line 391. Aedes (Stegomyia) aegypti: Stegomyia in italics; it is a subgeneric name!!!!!

Answer: ok, adjusted

Reviewer1: Line 411. Change "Neotropical Culioidae" to "Neotropical Culicidae".

Answer: ok, adjusted

Reviewer#2

Reviewer2: 1- Introduction - "Rural settlements are established in places that are favorable for human-vector interaction due to their proximity to extensive forest areas and the lack of adequate infrastructure for the residents who live there." I partially disagree with authors about their argument. The places are favorable or become favorable because of anthropogenic changes in the forest environments?

Answer: ok, adjusted

Reviewer2: What do authors mean by "the habitats of residents? Why habitat? Can it be socioecological conditions?

Answer: ok, adjusted

Reviewer2: Authors need to adjust the use of some words. For instance, it is not common to say "Mosquitoes are characterized by their flexible use of a wide variety of breeding

sites," The word flexible is not adequate, there are better word to use. i.e. generalist. Breeding sites can be replaced by larval habitats, adult habitat.

Answer: ok, adjusted Reviewer2: Please consider a better way to mention "from larges found at the soil level", ground pools, standing water, others? Authors must check the terminology used in mosquito ecology.

Answer: ok, adjusted Reviewer2: Authors must consider to use the correct ecological term for "small pools of water stored in leaves in the canopy of trees." Can it be phytotelma?

Answer: ok, adjusted

Reviewer2: 6- Immature forms of Anopheles species have been recorded in artificial containers [16,17], including An. darlingi Root, which was found in an artificial lagoon in the urban region of Manaus in the state of Amazonas. Please rewrite this part, because an artificial lagoon is not an artificial container.

Answer: ok, adjusted

Reviewer2: In the last paragraph of the introduction, authors say "Due to the importance of sanitary conditions and the ecological evolution that has led to the use of water containers for breeding by female mosquitoes, this work aimed to identify mosquito species that colonize different types of artificial breeding places, located in environments with different levels of anthropization within the state of Amazonas.

This entire paragraph needs to be rewritten. What do authors mean when they say sanitary conditions and ecological evolution? What is a water container? Why breeding by females? Can female breed on water containers? Can you replace artificial breeding

places by larval habitats? Because your investigation is limited to a single settlement in Presidente Figueiredo, I consider it is incorrect to say, "within the state of Amazonas". Your sample is small, and it is does not represent the immense diversity found in the Amazon. Please considering the focus of your study, rewrite this part.

Answer: ok, adjusted

Reviewer2: Can you please provide better details of the climate in the region studied?

Answer: ok, adjusted

Reviwer2: Please clarify your procedure to choose locations to install your traps. Did you used satellite images to select locations within the settlements area? How did you select locations within the settlement? You said you installed 9 traps per sampling point, please define what you mean by sampling point. Please, add this information, they are essential for an ecological study.

Answer: ok, adjusted

Reviewer2: Considering you used tap water to fill the traps, do you think your sampling was biased based on the female choice that depends on several molecules present in the habitats, including pheromones. How the water characteristics may have affect your sampling diversity and abundance and occurrence of mosquitoes?

Answer: ok, adjusted

Reviewer2: What do you mean by material collection? Please clarify.

Answer: ok, adjusted

Reviewer2: How did you avoided cross contamination among traps?

Answer: The larvitraps were installed and they were not removed until the end of all collection period. Thus, there was not risk of contamination among traps.

Reviewer2: I assume you installed 162 traps each field collection. Did you install all traps in a single day?

Answer: ok, adjusted

Reviewer2: How did you define the 15 days interval between installation and collection days?

Answer: ok, adjusted

Reviewer2: A believe the larvae were in early stages and most have died during the trip and in the lab. Have you counted and identified all of them? This needs to be clarified because of your statistical analyses and diversity indexes you used.

Answer: ok, adjusted

Reviewer2: Figure 3 legend needs to be together with the graphic. Answer: according to author guidelines, “Figure captions must be inserted in the text of the manuscript, immediately following the paragraph in which the figure is first cited. Do not include captions as part of the figure files themselves or submit them in a separate document.”.

Reviewer2: It is of paramount importance to explain why authors consider that the larvitraps were good to sample the most common mosquito species in the Rio Pardo agrovillage. Can you please provide reference (in addition to the thesis you cited) of other published studies on the same agrovillage? How can you support your argument that traps recovered the most common species? Please add reference. How can you be sure your trap collections in artificial habitats were able to return most species present in the region?

Answer: ok, adjusted

---

## [Editor Report · Decision Letter 1]

23 Sep 2020

Diversity of mosquitoes (Diptera: Culicidae) collected in different types of larvitraps in an Amazon rural settlement

PONE-D-20-18942R1

Dear Dr. Almeida,

We’re pleased to inform you that your manuscript has been judged scientifically suitable for publication and will be formally accepted for publication once it meets all outstanding technical requirements.

Kind regards,

Rafael Maciel-de-Freitas

Academic Editor

PLOS ONE

---

## [Editor Report · Acceptance letter]

25 Sep 2020

PONE-D-20-18942R1 

Diversity of mosquitoes (Diptera: Culicidae) collected in different types of larvitraps in an Amazon rural settlement 

Dear Dr. Almeida:

I'm pleased to inform you that your manuscript has been deemed suitable for publication in PLOS ONE. Congratulations! Your manuscript is now with our production department. 

Kind regards, 

on behalf of

Dr. Rafael Maciel-de-Freitas 

Academic Editor

PLOS ONE